# Self-Adversarial Learning with Comparative Discrimination for Text Generation

**Wangchunshu Zhou**[1]*    **Tao Ge**[2]    **Ke Xu**[1]    **Furu Wei**[2]    **Ming Zhou**[2]

[1]Beihang University, Beijing, China
[2]Microsoft Research Asia, Beijing, China

`zhouwangchunshu@buaa.edu.cn, kexu@nlsde.buaa.edu.cn`
`{tage, fuwei, mingzhou}@microsoft.com`

## Abstract

Conventional Generative Adversarial Networks (GANs) for text generation tend to have issues of reward sparsity and mode collapse that affect the quality and diversity of generated samples. To address the issues, we propose a novel self-adversarial learning (SAL) paradigm for improving GANs' performance in text generation. In contrast to standard GANs that use a binary classifier as its discriminator to predict whether a sample is real or generated, SAL employs a comparative discriminator which is a pairwise classifier for comparing the text quality between a pair of samples. During training, SAL rewards the generator when its currently generated sentence is found to be better than its previously generated samples. This self-improvement reward mechanism allows the model to receive credits more easily and avoid collapsing towards the limited number of real samples, which not only helps alleviate the reward sparsity issue but also reduces the risk of mode collapse. Experiments on text generation benchmark datasets show that our proposed approach substantially improves both the quality and the diversity, and yields more stable performance compared to the previous GANs for text generation.

## 1 Introduction

Generative Adversarial Networks (Goodfellow et al., 2014) (GANs) have achieved tremendous success for image generation and received much attention in computer vision. For text generation, however, the performance of GANs is severely limited due to reward sparsity and mode collapse: reward sparsity refers to the difficulty for the generator to receive reward signals when its generated samples can hardly fool the discriminator that is much easier to train; while mode collapse refers to the phenomenon that the generator only learns limited patterns from the real data. As a result, both the quality and diversity of generated text samples are limited.

To address the above issues, we propose a novel self-adversarial learning (SAL) paradigm for improving adversarial text generation. In contrast to standard GANs (Figure 1(a)) that use a binary classifier as its discriminator to predict whether a sample is real or generated, SAL employs a comparative discriminator which is a pairwise classifier assessing whether the currently generated sample is better than its previously generated one, as shown in Figure 1(b). During training, SAL rewards the generator when its currently generated samples are found to be better than its previously generated samples. In the earlier training stage when the quality of generated samples is far below the real data, this self-improvement reward mechanism makes it easier for the generator to receive non-sparse rewards with informative learning signals, effectively alleviating the reward sparsity issue; while in the later training stage, SAL can prevent a sample from keeping receiving high reward as the self-improvement for a popular mode will become more and more difficult, and therefore help the generator avoid collapsing toward the limited patterns of real data.

We comprehensively evaluate the proposed self-adversarial learning paradigm in both synthetic data and real data on the text generation benchmark platform (Zhu et al., 2018). Compared to the previous approaches for adversarial text generation (Yu et al., 2017; Che et al., 2017; Lin et al., 2017), our

---

*  This work was done during the first author's internship at Microsoft Research Asia.

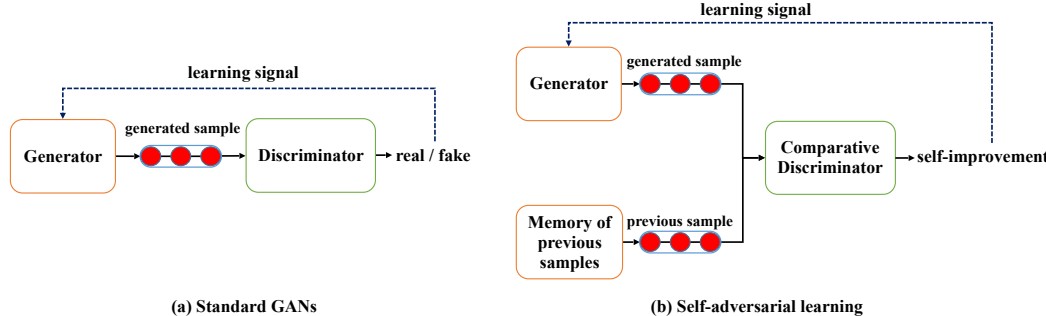

Figure 1: (a) Conventional adversarial learning that uses a binary real/fake classifier as its discriminator; (b): Self-adversarial learning that employs a comparative discriminator to compare the currently generated sample to its previously generated samples for obtaining rewards through self-improvement.

approach shows a substantial improvement in terms of both the quality and the diversity of generated samples as well as better performance stability in adversarial learning.

## 2 BACKGROUND: ADVERSARIAL TEXT GENERATION

Adversarial text generation has drawn much attention in recent years due to its advantages (e.g., sequence-level guidance without the exposure bias issue (Bengio et al., 2015)) over maximum likelihood estimation (MLE) for natural language generation. It formulates the learning process as a minimax game between a generator $G_\theta$ parameterized by $\theta$ and a discriminator $D_\phi$ parameterized by $\phi$: the discriminator is trained to distinguish between the samples drawn from the real data distribution $p_{data}$ and the samples generated by the generator; while the generator is trained to generate samples that can "fool" the discriminator. The adversarial learning objective of the generator and the discriminator can be formulated as:

$$\min_\theta \max_\phi \mathbb{E}_{\boldsymbol{x} \sim p_{\text{data}}}[\log D_\phi(\boldsymbol{x})] + \mathbb{E}_{\boldsymbol{z} \sim p_{\boldsymbol{z}}}[\log(1 - D_\phi(G_\theta(\boldsymbol{z})))] \tag{1}$$

where $\boldsymbol{x}$ is a sample from the real data, $G_\theta(\boldsymbol{z})$ is a sample generated by the generator with the initialization $\boldsymbol{z}$ that is drawn from the noise distribution $p_z$ (e.g., standard normal distribution).

While GANs have shown some promising results (Yu et al., 2017; Guo et al., 2018), there are two fundamental issues that impede their progress in text generation: (i) Reward sparsity, which is due to the fact that the discriminator tends to learn much better than the generator and thus easily recognizes generated samples as fake. In such cases, it will be difficult for the generator to receive rewards; (ii) Mode collapse, which arises from the intrinsic nature of GANs and leads the adversarial models to only learn the limited patterns from the real samples. These two issues limit the ability of GANs to generate high-quality and diverse text samples, which have not been well addressed yet.

## 3 SELF-ADVERSARIAL LEARNING

To address the aforementioned issues, we propose a novel self-adversarial learning (SAL) paradigm. Inspired by self-play (Silver et al., 2017; Rennie et al., 2017) in reinforcement learning, the core idea of SAL is to reward the generator if its currently generated sample is found to be better than its previously generated ones. Like AlphaGo (Silver et al., 2017), the generator in SAL struggles to learn to generate better samples than its previously generated samples for passing the "self-improvement" test by a comparative discriminator, which is a pairwise classifier trained to compare the quality of two samples, as Figure 1(b) shows.

Compared to conventional GANs (Figure 1(a)), SAL has the following advantages: First, in the earlier training stage when the quality of generated samples is far below the real data, the self-improvement

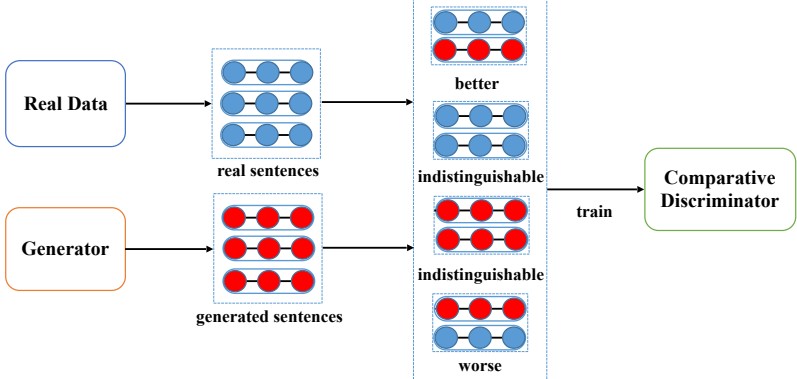

Figure 2: The training process of the comparative discriminator.

reward mechanism of SAL allows the generator to receive informative learning signals more easily as it makes the assessment of sample quality better adapted to the current capability of the generator, making it less likely to suffer from the issue of reward sparsity; Second, in the later training stage when the generated samples' quality is high, SAL can prevent a sample from keeping receiving high reward as it will become more and more difficult to pass the "self-improvement" test, thus reducing the risk of the generator collapsing towards limited patterns. The self-improvement mechanism and the 'tie' option in the comparative discriminator also provides a reasonable baseline which corresponds to cases where newly generated samples are found to be indistinguishable with previous ones, thus improving the training stability. We provide a more detailed qualitative analysis of why the proposed self-adversarial learning paradigm can alleviate these problems in Appendix.

## 3.1 COMPARATIVE DISCRIMINATOR

As introduced above, the core component for SAL is the comparative discriminator, which is a pairwise classifier comparing the quality of two samples. It learns a total order of sample quality and encodes the inductive bias that one sample is better ($>$), worse ($<$), or indistinguishable ($\approx$) in terms of its quality compared to the other. For a (text) sample, the comparative discriminator can offer more informative learning signals than the conventional binary (i.e., real/fake) classification based discriminator because the sample can receive multiple feedbacks from the comparative discriminator by comparing it with multiple other samples.

For training the comparative discriminator, we construct pairwise training examples from the real and generated samples, as Figure 2 shows. For a real sample $s_+$ and a generated sample $s_-$, we assign the label "better ($>$)" to the pair ($s_+$, $s_-$) and "worse ($<$)" to ($s_-$, $s_+$). For two samples both from real data or from the generated samples, we assign the label "indistinguishable ($\approx$)" to such pairs (i.e., ($s_+^i$, $s_+^j$) and ($s_-^i$, $s_-^j$)). For a training set with $n$ real samples and $n$ generated samples, the comparative discrimination can construct $\binom{2n}{2}$ pairwise training examples, allowing to enhance the generalization ability of the comparative discriminator.

Moreover, to improve the model's ability to distinguish between good generated samples and bad generated samples for self-play learning, we additionally select the samples generated during the later stage of MLE training as pseudo-real samples, and select those generated in the earlier epochs when the generator does not fully converge as fake sentences. We then pair the pseudo-real samples with the fake samples to construct training instances to supervise the model to compare their qualities. In this way, the comparative discriminator is prevented from being taught to always recognize two generated samples as equally bad and assign zero reward to the generator. As a result, it can become more sensitive to the quality difference in a pair of text samples and thus allow the generator to receive rewards more easily.

## 3.2 TRAINING

Before we formally introduce the training procedure for SAL, we first define the learning objective for the comparative discriminator $D_\phi$ and the generator $G_\theta$ in SAL:

$$L_D = -\mathbb{E}_{(\boldsymbol{x}_1,\boldsymbol{x}_2)\sim(\mathcal{M}\cup p_{\text{data}}(x))^2}[\log D_\phi^{Q(\boldsymbol{x}_1,\boldsymbol{x}_2)}(\boldsymbol{x}_1,\boldsymbol{x}_2)] \tag{2}$$

$$L_G = -\mathbb{E}_{(\boldsymbol{z},\boldsymbol{x}_r)\sim(p_z(\boldsymbol{z}),\mathcal{M})}\Big[\sum_{q\in\{>,<,\approx\}} w_q \log D_\phi^q(G_\theta(\boldsymbol{z}),\boldsymbol{x}_r)\Big] \tag{3}$$

In Eq (2) and Eq (3), $\mathcal{M}$ is the set of previous generated samples by the generator, $Q(\boldsymbol{x}_1,\boldsymbol{x}_2) \in \{>,<,\approx\}$ is the true label for the pair $(\boldsymbol{x}_1, \boldsymbol{x}_2)$, $D_\phi^q(\boldsymbol{x}_1,\boldsymbol{x}_2)$ is the probability of the comparative discriminator's prediction being $q$ ($q \in \{>,<,\approx\}$) for the pair $(\boldsymbol{x}_1, \boldsymbol{x}_2)$. $w_q$ is the reward weight for the case $q$, which is a hyperparameter for SAL. If the generator generates a sample $G_\theta(\boldsymbol{z})$ that is better ($>$) than its previously generated sample $\boldsymbol{x}_r$, it receives a positive reward; if $G_\theta(\boldsymbol{z})$ is worse ($<$) than $\boldsymbol{x}_r$, it receives a negative reward; while if the quality of $G_\theta(\boldsymbol{z})$ is classified as similar ($\approx$) to $\boldsymbol{x}_r$, it receives zero credit. Therefore, we have $w_{(>)} > 0 = w_{(\approx)} > w_{(<)}$.

Since $L_G$ can only be directly optimized in standard continuous GAN training, we alternatively employ the policy gradient algorithm (Sutton et al., 2000) to train the generator, as previous approaches for adversarial text generation. For SAL, we define the reward for a generated sample $\boldsymbol{x}_g$ compared with a reference sample $\boldsymbol{x}_r$ which is a previous generated sample by the generator as the weighted reward based on the probability distribution of the prediction by the comparative discriminator:

$$\gamma_\phi(\boldsymbol{x}_g,\boldsymbol{x}_r) = w_{(>)}D_\phi^{(>)}(\boldsymbol{x}_g,\boldsymbol{x}_r) + w_{(<)}D_\phi^{(<)}(\boldsymbol{x}_g,\boldsymbol{x}_r) + w_{(\approx)}D_\phi^{(\approx)}(\boldsymbol{x}_g,\boldsymbol{x}_r) \tag{4}$$

In text generation, the generator $G_\theta$ obtains the reward only when one sample has been completely generated, which means no intermediate reward is gained. To relieve this problem, following the practice in SeqGAN (Yu et al., 2017), we utilize the Monte Carlo rollout method to approximate intermediate rewards by sampling unknown tokens with generated prefix $Y_{1:t}$ following generator policy $G_\theta$ till sample completion. Empirically, we found that the Monte Carlo rollout also helps to reduce the variance of the reference sample utilized for comparison. We calculate the expected reward as

$$\mathcal{R}_{\theta,\phi}(s = Y_{1:t-1}, a = y_t) = \mathbb{E}_{(\boldsymbol{x}_g,\boldsymbol{x}_r)\sim(G_\theta(Y_{1:t-1}),\mathcal{M})}[\gamma_\phi(\boldsymbol{x}_g,\boldsymbol{x}_r)] \tag{5}$$

The objective of the generator is to generate a sequence to maximize its expected final reward. With likelihood ratio (Sutton et al., 2000), we can formulate the gradient of the objective function for generator $G_\theta$ as:

$$\nabla_\theta J(\theta) = \sum_{t=1}^{T} \mathbb{E}_{Y_{1:t-1}\sim G_\theta}[\nabla_\theta \log(G_\theta(y_t|Y_{1:t})) \cdot \mathcal{R}_{\theta,\phi}(s = Y_{1:t-1}, a = y_t)] \tag{6}$$

To improve the training of self-adversarial learning, we borrow ideas from the field of deep reinforcement learning and propose two training techniques to improve self-adversarial training.

**Scheduled rewarding** Similar to the exploitation-exploration trade-off in reinforcement learning (Langford & Zhang, 2007), the positive reward assigned for actions generating better samples encourage exploration while the penalty for generating worse samples makes the generator more conservative. Intuitively, in the earlier stage of self-adversarial learning, the generator should explore better policy by receiving higher rewards for relative progress; while in the later stage, the generator should be more conservative by penalizing more for worse samples to avoid performance degradation. We simply decrease $w_{(>)}$ and increase $w_{(<)}$ linearly with training iteration and refer this technique as scheduled rewarding.

---

**Algorithm 1** Self-Adversarial Learning With Comparative Discriminator

---

**Require:** Generator $G_\theta$; comparative discriminator $D_\phi$; samples of real sentences $S_+$; self-adversarial learning step $g$; discriminator step $k$; memory buffer $\mathcal{M}$ for the previous generated samples
1: Pretrain $G_\theta$ using MLE on $S_+$
2: Generate samples with $G_\theta$ and store them into $\mathcal{M}$
3: **repeat**
4:   **for** $k$ steps **do**
5:     Collect a mini-batch of balanced sample pairs $(\boldsymbol{x}_1, \boldsymbol{x}_2)$ from $\mathcal{M} \cup S_+$
6:     Update $D_\phi$ via Eq (2)
7:   **end for**
8:   **for** $g$ steps **do**
9:     Generate a mini-batch of samples $\boldsymbol{x}_g \sim G_\theta$
10:     Collect a mini-batch of reference samples $\boldsymbol{x}_r$ from $\mathcal{M}$
11:     Update $G_\theta$ via Eq (6)
12:   **end for**
13:   Update $\mathcal{M}$ with $G_\theta$
14: **until** Convergence

---

**Memory replay** Continuously comparing the generator with its most recent stage may suffer from the correlation between generated samples and reference samples, which makes the training process unstable. Inspired by experience replay (Lillicrap et al., 2015), we construct a memory buffer which contains samples generated in the last $K$ training steps. Reference samples are sampled from the memory buffer rather than samples generated in the most recent stage of the generator, which empirically helps stabilize the training process.

The training process of SAL is summarized in Algorithm 3. Self-adversarial learning with the proposed comparative discriminator achieves Nash Equilibrium when the generator models the distribution of real samples perfectly. In this case, the comparative discriminator cannot successfully distinguish generated samples from real samples and tends to recognize two samples as "indistinguishable". The reward received by the generator is thus zero and training converges. However, how a non-Bernoulli GAN converges to such an equilibrium is still an open problem (Goodfellow et al., 2014; Goodfellow, 2014) and is beyond the scope of this work.

## 4 EXPERIMENTS

### 4.1 EXPERIMENTAL SETTING

Following the experimental settings in previous work (Lin et al., 2017; Guo et al., 2018; Shi et al., 2018; Zhu et al., 2018; Nie et al., 2018), we evaluate our approach in both synthetic and real datasets based on Texygen (Zhu et al., 2018), which is a benchmark platform for evaluating adversarial text generation models. Table 1 presents the basic information of the datasets used for evaluation.

Table 1: Description of the datasets used for evaluation

|  | Synthetic | Image COCO | EMNLP2017 WMT NEWS |
|---|---|---|---|
| Category | simulated data | image description | news article |
| Vocabulary size | 5000 | 4682 | 5255 |
| Sequence length | 20/40 | <37 | <51 |
| Sentence number (training) | 10000 | 10000 | 270000 |
| Sentence number (test) | 10000 | 10000 | 10000 |

As SeqGAN (Yu et al., 2017), our generator is a single-layer LSTM (Hochreiter & Schmidhuber, 1997) and our discriminator is almost based on TextCNN (Kim, 2014) except that it concatenates the feature representation of two compared samples and outputs the probability for their comparative relations (i.e., $>, <, \approx$). We keep the most of the hyperparameters same with the SeqGAN except the hyperparameters introduced by our models (i.e., $w_{(>)}, w_{(<)}, w_{(\approx)}$) which are tuned based on the synthetic experiment and kept the same for the real data experiments.

We evaluate the adversarial text generation models in terms of both quality and diversity. Following the prior work, in the synthetic dataset, we use the oracle LSTM to evaluate the negative log-likelihood

Table 2: Performance comparison of different models in synthetic tests where sequence length is set to 20 and 40 respectively. For all the metrics presented, the lower, the better.

| Method | $NLL_{oracle}(20/40)$ | $NLL_{gen}(20/40)$ | $NLL_{oracle} + NLL_{gen}(20/40)$ |
|---|---|---|---|
| SAL | **7.71** ±0.17 / **9.31**±0.03 | **6.58**±0.15 / **6.97** ±0.05 | **14.29** ±0.11 / **16.24**±0.03 |
| SeqGAN | 8.63 ±0.19 / 9.63 ±0.04 | 6.61±0.22 / 6.98±0.08 | 15.00±0.03 / 16.35±0.02 |
| RankGAN | 8.42 ±0.31 / 9.52±0.11 | 7.14±0.34 / 7.05±0.12 | 15.01±0.02 / 16.37±0.02 |
| MaliGAN | 8.74 ±0.16 / 9.67 ±0.03 | 6.62±0.25 / 7.14±0.09 | 15.03±0.03 / 16.39±0.03 |
| MLE | 9.05 ±0.03 / 9.84±0.02 | **5.96**±0.02 / **6.55**±0.02 | 15.02±0.03 / 16.39±0.01 |

of our generated samples (denoted as $NLL_{oracle}$) as the metric to reflect quality; for the diversity metric, we use the negative log-likelihood of the synthetic dataset (denoted as $NLL_{gen}$) evaluated by the generator with the best quality (i.e., the best $NLL_{oracle}$ score) during training. We also use the best $NLL_{oracle}$ + $NLL_{gen}$ obtained during training to evaluate the quality-diversity trade-off.

For the real data experiments, we follow the previous work to apply the commonly-used BLEU scores (Papineni et al., 2002) (BLEU(F)) and the perplexity of generated samples evaluated by an open-sourced pretrained language model (Jozefowicz et al., 2016) as quality metrics since $NLL_{oracle}$ cannot be evaluated without an oracle language model. For evaluating diversity, we employ both backward BLEU (Shi et al., 2018) (BLEU(B)) which evaluates the test data using generated samples as reference and $NLL_{gen}$ as the metrics. To evaluate the generated samples in more aspects, we calculate frechet distance (Heusel et al., 2017) (FD) between generated samples and real data with sentence representation obtained by InferSent (Conneau et al., 2017) which is a pre-trained sentence embedding model.

We compare our approach to previous well-known adversarial text generation models including SeqGAN (Yu et al., 2017), RankGAN (Li et al., 2017) and MaliGAN (Che et al., 2017). Leak-GAN (Guo et al., 2018) and RelGAN (Nie et al., 2018) focus on architecture-level modification, which is orthogonal to our work and the proposed self-adversarial learning paradigm can be applied to them as well. We provide results of the combination of LeakGAN with SAL in the Appendix.

In the following sections, we denote our proposed self-adversarial learning approach as SAL. Since adversarial training is very sensitive to random initialization and suffers from high variance, we conduct five individual runs with different random seeds for each model and report the mean and the standard deviation of the obtained results.

## 4.2 EXPERIMENTAL RESULTS

### 4.2.1 RESULTS IN SYNTHETIC DATA

Table 2 shows the results in the synthetic dataset. We can observe that SAL largely outperforms the previous GANs in all metrics in both cases of sequence length 20 and 40. Although its performance in $NLL_{gen}$ is worse than MLE as MLE directly optimizes the metric, it yields better quality-diversity trade-off than MLE training, which has not been achieved by the previous GANs, which is shown by the fact that the $NLL_{oracle}$+$NLL_{gen}$ for SAL is lower than that yielded by MLE, while other GANs have the same sum score with MLE, indicating that they fail to improve the quality-diversity trade-off after pretraining. This demonstrates SAL's advantage in alleviating the mode collapse problem. We also find that the training of SAL is more stable compared with other text GANs.

In addition, we find SAL learns faster and better than the other GANs by comparing their performance curves of $NLL_{oracle}$ during training in Figure 3, which is consistent with our intuition that the self-improvement reward mechanism in SAL can alleviate the reward sparsity issue in the earlier training stage and help the generator learn better.

### 4.2.2 RESULTS IN REAL DATA

The results for COCO image caption dataset are presented in Table 3 and the performance curve of the perplexity during training is shown in Figure 4. As in the synthetic data, we observe that our SAL consistently yields better results in all the metrics with stable performance (i.e., low variance) compared to the previous GANs. According to Table 18 and Figure 4, SeqGAN and our SAL can improve MLE in the quality metrics (i.e., BLEU (F) and Perplexity) while MaliGAN and RankGAN

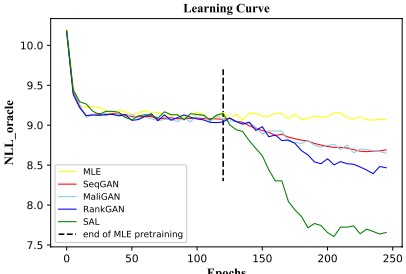 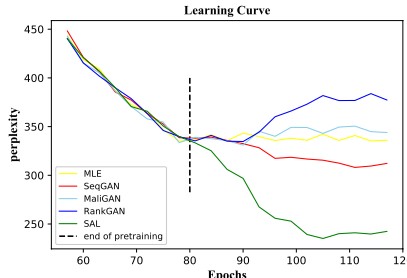

Figure 3: The performance curves of NLL-oracle during training in the synthetic dataset

Figure 4: The performance curves of perplexity during training in the Image COCO dataset

Table 3: Performance comparison of different models in the COCO caption generation task. Metrics from top to bottom represent respectively the generation quality, the generation diversity, and the divergence between real data of generated sentences. For all the BLEU metrics, the higher, the better; for $NLL_{gen}$ and FD, the lower, the better.

| Metrics | MLE | SeqGAN | MaliGAN | RankGAN | SAL |
|---------|-----|--------|---------|---------|-----|
| BLEU-2(F) | 0.730 ±0.01 | 0.748 ±0.03 | 0.733 ±0.03 | 0.727 ±0.04 | **0.785** ±0.02 |
| BLEU-3(F) | 0.494 ±0.01 | 0.514 ±0.04 | 0.497 ±0.04 | 0.491 ±0.04 | **0.581** ±0.03 |
| BLEU-4(F) | 0.303 ±0.01 | 0.307 ±0.02 | 0.295 ±0.03 | 0.291 ±0.03 | **0.362** ±0.02 |
| BLEU-5(F) | 0.187 ±0.01 | 0.187 ±0.02 | 0.178 ±0.02 | 0.175 ±0.03 | **0.227** ±0.02 |
| Perplexity | 338.4 ±7.6 | 307.2 ±14.9 | 343.8 ±21.3 | 391.2 ±35.1 | **231.3** ±10.8 |
| BLEU-2(B) | **0.759** ±0.02 | 0.694 ±0.03 | 0.676±0.03 | 0.683±0.04 | 0.724 ±0.02 |
| BLEU-3(B) | **0.531** ±0.02 | 0.472±0.03 | 0.443±0.03 | 0.449±0.04 | 0.503 ±0.03 |
| BLEU-4(B) | **0.332** ±0.02 | 0.285±0.02 | 0.279±0.02 | 0.277±0.02 | 0.313±0.02 |
| BLEU-5(B) | **0.209** ±0.02 | 0.186±0.02 | 0.178±0.02 | 0.182 ±0.02 | 0.198 ±0.02 |
| $NLL_{gen}$ | **0.721** ±0.02 | 1.035 ±0.02 | 1.052 ±0.02 | 1.145 ±0.02 | 0.873 ±0.02 |
| FD | **0.043** ±0.009 | 0.065 ±0.018 | 0.076 ±0.021 | 0.083 ±0.027 | **0.051** ±0.014 |

Table 4: Performance comparison of different models in the EMNLP2017 WMT news generation task. Metrics from top to bottom represent respectively the generation quality, the generation diversity, and the divergence between real and generated data. For all the BLEU metrics, the higher, the better. For $NLL_{gen}$ and FD, the lower, the better.

| Metrics | MLE | SeqGAN | MaliGAN | RankGAN | SAL |
|---------|-----|--------|---------|---------|-----|
| BLEU-2(F) | 0.769 ±0.02 | 0.761 ±0.03 | 0.764 ±0.03 | 0.736 ±0.02 | **0.788** ±0.02 |
| BLEU-3(F) | 0.475 ±0.01 | 0.463 ±0.03 | 0.468 ±0.03 | 0.441 ±0.04 | **0.523** ±0.02 |
| BLEU-4(F) | 0.243 ±0.02 | 0.228 ±0.03 | 0.231 ±0.03 | 0.204 ±0.02 | **0.281** ±0.02 |
| BLEU-5(F) | 0.124 ±0.02 | 0.115 ±0.02 | 0.113 ±0.03 | 0.095 ±0.02 | **0.149** ±0.02 |
| Perplexity | 702.6 ±18.6 | 743.8 ±34.2 | 825.1 ±44.3 | 975.4 ±65.1 | **612.8** ±22.5 |
| BLEU-2(B) | **0.741** ±0.02 | 0.693 ±0.03 | 0.684±0.03 | 0.671±0.03 | **0.726**±0.02 |
| BLEU-3(B) | **0.476**±0.01 | 0.413±0.03 | 0.391±0.04 | 0.373±0.05 | 0.431±0.03 |
| BLEU-4(B) | **0.245**±0.01 | 0.216±0.03 | 0.197±0.03 | 0.191±0.03 | 0.232±0.02 |
| BLEU-5(B) | **0.129**±0.01 | 0.112 ±0.02 | 0.094±0.02 | 0.096±0.03 | 0.123±0.02 |
| $NLL_{gen}$ | **2.386**±0.01 | 2.732±0.04 | 2.862±0.06 | 3.157±0.11 | 2.578±0.04 |
| FD | **0.079** ±0.012 | 0.172 ±0.032 | 0.194 ±0.043 | 0.219 ±0.052 | **0.137** ±0.023 |

perform comparably to MLE. However, in the WMT NEWS dataset where text samples tend to be longer, we observe something different from Table 4: all the previous GANs fail to improve MLE. This is because long text generation makes the discrepancy between generated samples and real samples very large even after MLE pre-training. As a result, previous GANs fail to stably enhance the sample quality due to the reward sparsity issue. In contrast, our SAL consistently performs well and largely improves the quality metrics over MLE. In addition, we observe that the diversity of samples generated by SAL is much better than previous GANs and only marginally worse than MLE, indicating SAL is helpful in reducing the risk of mode collapse.

In addition to the automatic metrics, we also conduct human evaluation for the generated samples. As previous work (Nie et al., 2018), we randomly sample 20 sentences from each model and pool them

Table 5: Human evaluation results of different models in both datasets. Scores are between 1-5, higher score indicates better quality.

| Dataset | MLE | SeqGAN | MaliGAN | RankGAN | SAL |
|---------|-----|--------|---------|---------|-----|
| COCO | 2.96±0.51 | 3.26±0.56 | 3.14±0.57 | 2.91±0.62 | **3.84±0.56 (p<=0.01)** |
| WMT NEWS | 2.35±0.86 | 2.19±0.88 | 2.24±0.87 | 2.05±0.91 | **2.65±0.89 (p<=0.01)** |

Table 6: Results of the ablation tests in the synthetic data and the COCO dataset.

| Dataset | SAL | CAL | w/o comparative | w/o "≈" | w/o scheduled rewarding | w/o memory replay |
|---------|-----|-----|-----------------|---------|-------------------------|-------------------|
| Synthetic (NLL) | 14.29 ±0.11 | 14.65 ±0.16 | 15.01 ±0.04 | 14.85 ±0.16 | 14.46±0.18 | 14.41 ±0.17 |
| COCO (Perplexity) | 231.3±10.8 | 276.7±12.5 | 341.8 ±13.4 | 291.5 ±16.3 | 248.3 ±14.2 | 254.6 ±13.7 |

with anonymizing the models' identity. We invite 20 graduate students with good English proficiency to score each sentence on a scale of 1-5 in terms of quality. According to Table 5, our SAL is consistently well rated in the human evaluation and outperforms all the baselines in both COCO and WMT NEWS datasets. We perform the Wilcoxon Rank Sum Test with the human evaluation results and find that samples generated by baseline models can be distinguished from samples generated by SAL with $p < 0.01$. Details of human evaluation procedure and samples generated by compared methods in two real-world datasets are presented in the Appendix.

## 4.3 DISCUSSION

To better understand SAL, we perform multiple ablation tests in both the synthetic and the real data. We employ $\text{NLL}_{oracle} + \text{NLL}_{gen}$ score with sequence length 20 as the evaluation metric for the synthetic data, denoted as NLL. For the real data, we use the perplexity of generated samples trained with COCO dataset as the evaluation metric. We compare SAL with the following reduced models:

- *CAL*: Replacing the comparison between the generated samples (i.e., self-play) to the comparison between the real and generated samples.

- *w/o comparative*: Using the binary discrimination scores of other generated samples as baseline for the policy gradient algorithm, which can be considered as a combination of the self-critic training (Rennie et al., 2017) with RL-based text GANs.

- *w/o "≈"*: Replace the three-class comparative discriminator with a binary comparative discriminator by removing the "≈" class.

- *w/o scheduled rewarding* and *w/o memory replay*

The results of the ablation tests are shown in Table 6. By observing the improvement by SAL over CAL, we confirm the importance of the self-play paradigm in SAL. It is notable that the proposed comparative discriminator alone (i.e., CAL) can yield good performance, demonstrating the effectiveness of learning by comparison. When replacing the comparative discriminator with the naive combination of self-critic baseline with text GANs, the performance largely decreases because the reward sparsity issue will be intensified when subtracting two already sparse rewards, this motivates the proposed pairwise comparative discriminator which makes self-comparison possible.

In addition, we find that the "≈" option plays a critical role in improving the result, without which the performance degrades significantly because it makes the task less trivial and provides a baseline for the policy gradient algorithm. Moreover, the training techniques (i.e., scheduled rewarding and memory replay) borrowed from deep reinforcement learning are also shown useful in improving the results but not so important as the core components (e.g., self-play and the comparative discriminator).

## 5 RELATED WORK

Many variants of GANs (including TextGAN (Zhang et al., 2017), GSGAN (Kusner & Hernández-Lobato, 2016), SeqGAN (Yu et al., 2017), MaliGAN (Che et al., 2017), RankGAN (Lin et al., 2017), FMGAN (Chen et al., 2018), LeakGAN (Guo et al., 2018), and RelGAN (Nie et al., 2018)) have been proposed for text generation as adversarial training has received increasing attention in recent

years. Typically, they address the non-differentiable issue by making continuous approximation or reinforcement learning. These approaches introduce several different architectures and optimization objectives of both the generator and the discriminator for adversarial text generation. Among the previous studies for adversarial text generation, the most related work to ours is RankGAN (Lin et al., 2017) which proposes a ranker to replace the conventional binary classifier as its discriminator for allowing the discrimination process to involve richer information. Another work whose idea is similar to ours is the relativistic discriminator (Jolicoeur-Martineau, 2018) (RGAN). It compares binary scores assigned to generated samples and real samples by subtraction as the learning signal to implicitly represent the inductive bias that half of the samples received by the discriminator is fake. In contrast, our comparative discriminator directly encodes this inductive bias and assesses generated sentences by comparison with a pairwise classifier, which provides more informative learning signals than subtraction in RGAN (Jolicoeur-Martineau, 2018) and normalized feature similarity in RankGAN (Lin et al., 2017). Our work is also related to the concurrent work (Zhou & Xu, 2020) that learns a comparative evaluator to evaluate open-domain natural language generation models.

## 6 CONCLUSION AND FUTURE WORK

We present a self-adversarial learning (SAL) paradigm for adversarial text generation. SAL rewards the generator when its comparative discriminator finds the generator becomes better than before. Through the self-improvement reward mechanism, the problem of reward sparsity and mode collapse can be alleviated and training of text GANs are more stable, which results in a better performance in the text generation benchmarks in terms of both quality, diversity, and lower variance. In the future, we plan to generalize our approach to other domains and modals to explore the potential of SAL for adversarial learning. Generated samples are presented in the Appendix together with other details, including human evaluation details and qualitative analysis of the proposed SAL.

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

## A   GENERATED SAMPLES

We present sentences generated by our proposed model and compared models to provide qualitative evaluation of different adversarial text generation models. From the presented generated samples, we can observe that samples generated by MLE training are less realistic compared with other samples. SeqGAN yield slightly better sample quality but the loss of diversity is observable even within randomly sampled 15 sentences. Adversarial training with proposed comparator, when trained by comparing with real samples, yield better quality but still lack of diversity. Finally, with the proposed self-adversarial learning paradigm, both quality and diversity of generated samples are improved.

### A.1   GENERATED SAMPLES IN IMAGE COCO DATASET

Table 7: Samples generated by SAL in Image COCO dataset

| |
|---|
| a picture of a person 's umbrella in a cell phone . |
| a man stands in a green field . |
| a young boy riding a truck . |
| a man on a motorcycle is flying on a grassy field . |
| a girl on a motorcycle parked on a city street . |
| a motorcycle parked in a city street . |
| a group of bikers riding bikes on a city street . |
| a kitchen with a cat on the hood and a street . |
| a bathroom containing a toilet and a sink . |
| a young woman in a kitchen with a smiley face . |
| a jet plane on the side of a street . |
| a dish is sitting on a sidewalk next to a baby giraffe . |
| a dog on a large green bike parked outside of the motor bike . |
| a person on a kawasaki bike on a race track . |
| a commercial aircraft is parked in front of a kitchen . |

Table 8: Samples generated by CAL in Image COCO dataset

| |
|---|
| a man is on a towel on a table outside of a real kitchen . |
| a group of lambs at a tall building . |
| a young boy riding a truck . |
| a man on a motorcycle is flying on a grassy field . |
| a man with a computer desk next to a white car . |
| a cat is on the walls of a cat . |
| a plane on a runway with a plane . |
| an elegant , dilapidated plane are standing in front of a parking bag . |
| the woman is riding a bike on their way . |
| a man wearing an old bathroom with a banana . |
| a plane is taking off from the ground . |
| a man holding a man in front of herself . |
| a woman is walking across the road . |
| a kitchen with an island in green tiles . |
| a clean kitchen with two small appliances . |

Table 9: Samples generated by SeqGAN in Image COCO dataset

a large image of a herd of racing train .
man and woman on horse .
a plane on a runway with a plane .
a man preparing a table with wood lid .
a view , tiled floors and a man prepares food .
a man wearing an old bathroom with a banana .
a man is is with a camera .
two people are parked on a street .
a white and white black kitten eating on a table .
a toilet is lit on the walls .
a kitchen is taking off from a window .
a man is wearing glasses wearing scarf .
a kitchen with graffiti hanging off from an open plain .
two women playing with the orange .
a kitchen with an island in a clear glass .

Table 10: Samples generated by MLE in Image COCO dataset

a jet airplane flies flying through front from an airplane .
a furry tub and overhead pot .
a man in a kitchen filled with dark lights green side , ..
a cross baby field dressed making cardboard
a bathroom with a small tub and oven .
a man above a bathroom with an oven room .
a jet airliner flying through the sky .
a kitchen with a dishwasher , and plenty of pots , pans .
a person holding onto two red era arena sits on the street .
a bathroom with a toilet and a bath tub .
a cat perched on the phone and a baseball cap .
the view of the street filled with really parked at the gates on the road .
a large hairy dog on a high bike with a cake .
a man is riding a white back bench .
a narrow bed and white spotted dark tiled walls .

## A.2 GENERATED SAMPLES IN EMNLP2017 WMT DATASET

Table 11: Samples generated by SAL in EMNLP2017 WMT dataset

(1) it ' s likely to be egyptian and many of the canadian refugees , but for a decade .
(2) the ministry spokesperson also said it now significant connected to the mountain.
(3) it is the time they can more competitive , where we have another $ 99 . 100 per cent , and completely on the alternative , and that ' s being affected .
(4) we expect $ 200 and 0 . 3 percent for all you form other , and , which then well , it ' s done .
(5) so we wouldn ' t feel very large in the game , but you fail to fund , and and the paper that ' s like its start .
(6) other countries made a playoff cut with pages by mrs . trump ' s eighth consecutive season as a president .

Table 12: Samples generated by CAL in EMNLP2017 WMT dataset

(1) i didn ' t put relatively quiet , we have , ' his work right in the particular heat rate , take steps traditionally clean .
(2) why the u . s . then the table is our cabinet to do getting an vital company for the correct review .
(3) those had trained for that , but no thin percentage of the nhs about being warned about the palestinian election before obama is not connected in israel .
(4) in course , voters - obama said : " torture is the outcome , the most powerful trade - popularity is happening in it as a success .
(5) " in 2012 , it is nice to remain - no trump actor established this night - scoring three films .
(6) we kind of not listen to knowing my most one , only , for a really good vote , and where things fun , you know .

Table 13: Samples generated by SeqGAN in EMNLP2017 WMT dataset

(1) his missed 4 , 000 the first 95 really 69 - year - olds .
(2) but just things , you want to thank it as my playing side has begun meeting with " and " the score had to train up , so he was tied for 11 years .
(3) and when he got back doing fresh ties with his election , he will now step in january , back.
(4) when you ' t know if i saw her task to find himself more responsibility ago .
(5) his hold over - up to a nine hike in 2015 , 13 percent of recently under suspects dead day , 24 , and to the city .
(6) " i look up on by the city ' s vehicle on the day in a meeting in november .

Table 14: Samples generated by MLE in EMNLP2017 WMT dataset

(1) you know that that is great for our ability to make thinking about how you know and you ?
(2) when it ' s a real thing possible , is if you the first time in a time here and get .
(3) u . s , now government spending at the second half of four years , a country where the law will join the region to leave japan in germany .
(4) deputy president , the issue of government and geneva probe threats and not - backed trump , but well - changing violence for their islamic state militants were innocent people .
(5) he suggested in a presidential primary source and comment on its size following protests conducted by 18 , some in 2012 will be looked at tech energy hub .
(6) " it ' s growing heavy hard , " mr . romney said , he says matters that can ' t again become the asian player .

# B  CASE STUDY: WHY IT WORKS

In this section, we present several qualitative case study examples to illustrate why comparative discrimination and self-adversarial learning helps mitigate the problem of reward sparsity and mode collapse. We extract a typical sentence generated during the initial stage of adversarial learning: "a man holding a suitcase holding a phones." We see that this sentence is of limited quality and is easily recognized by binary discriminator in SeqGAN with high confidence, this makes the credit received by the generator very sparse and makes training difficult. Comparative adversarial learning (CAL) where we use the comparative discriminator to assess the quality of this sample by comparing it with a real sample helps as comparative discrimination have three catagories, which is less trivial. The improvement is not so much as the discrepancy of generated samples and real samples is fairly large. However, with proposed self-adversarial learning paradigm, the comparative discriminator assesses this sentence by comparing it with a previous generated sentence which is also of poor quality. The self-improvement is easier to be recognized by the comparative discriminator and makes this sample get good rewards.

As the comparative discriminator has to learn a total order of sample quality which is more challenging than standard binary discrimination, the chance of the comparative discriminator to be over-trained is reduced, which makes the model easier to achieve self-improvement, thus help to alleviate the reward sparsity problem.

We also extract a sentence generated in the late stages which is fairly good and fools the binary discriminator: "a woman sitting on a bench on a park." In standard adversarial text generation models, a sentence like this would keep receiving large rewards and result in mode collapse. In self-adversarial learning paradigm, this sentence is not much better than other sentences generated by the generator itself, so its reward is limited, which reduces the risk of mode collapse.

Table 15: Case study of comparative discrimination and self-adversarial learning.

| Generated sentence | Reference sentence | Reward |
|---|---|---|
| a man holding a suitcase holding a phones. | - (SeqGAN) | 0.018 |
| a man holding a suitcase holding a phones. | a student walks in the rain with a green umbrella. (Real) | 0.051 |
| a man holding a suitcase holding a phones. | a men 's kitchen and a cow. (Self) | 0.561 |
| a woman sitting on a bench on a park. | - (SeqGAN) | 0.825 |
| a woman sitting on a bench on a park. | a young man rides his bicycle on top of a cement bench. (Real) | 0.438 |
| a woman sitting on a bench on a park. | a man sitting on a table watching a television. (Self) | 0.086 |

## B.1  ABLATED MODELS

For the ablated model variants, SAL w/o self-play and w/o comparative discriminator is trained with the following algorithms. Specifically, the difference between SAL and CAL is that the reference sample which is compared with the currently generated sample is replaced by a real sample instead of a previously generated one. For the variant without the comparative discriminator, we employ a binary discriminator trained in the same way with the vanilla GAN, as for the reward of generating $x_g$, we first sample a previously generated sample $x_r$ as reference and calculate the reward as $D(x_g) - D(x_r)$.

---

**Algorithm 2** Self-Adversarial Learning without self-play (i.e. CAL)

---

**Require:** Generator $G_\theta$; comparative discriminator $D_\phi$; samples of real sentences $S_+$; self-adversarial learning step $g$; discriminator step $k$; memory buffer $\mathcal{M}$ for the previous generated samples
1: Pretrain $G_\theta$ using MLE on $S_+$
2: Generate samples with $G_\theta$ and store them into $\mathcal{M}$
3: **repeat**
4:    **for** $k$ steps **do**
5:       Collect a mini-batch of balanced sample pairs $(\boldsymbol{x}_1,\boldsymbol{x}_2)$ from $\mathcal{M} \cup S_+$
6:       Update $D_\phi$ via Eq (2)
7:    **end for**
8:    **for** $g$ steps **do**
9:       Generate a mini-batch of samples $\boldsymbol{x}_g \sim G_\theta$
10:      Collect a mini-batch of reference samples $\boldsymbol{x}_r$ from $\mathcal{S}_+$
11:      Update $G_\theta$ via Eq (6)
12:    **end for**
13:    Update $\mathcal{M}$ with $G_\theta$
14: **until** Convergence

---

**Algorithm 3** Self-Adversarial Learning without comparative discriminator

---

**Require:** Generator $G_\theta$; binary discriminator $D_\phi$; samples of real sentences $S_+$; self-adversarial learning step $g$; discriminator step $k$; memory buffer $\mathcal{M}$ for the previous generated samples
1: Pretrain $G_\theta$ using MLE on $S_+$
2: Generate samples with $G_\theta$ and store them into $\mathcal{M}$
3: **repeat**
4:    **for** $k$ steps **do**
5:       Collect a mini-batch of generated samples from $\mathcal{M}$.
6:       Update $D_\phi$ with conventional GAN discriminator loss
7:    **end for**
8:    **for** $g$ steps **do**
9:       Generate a mini-batch of samples $\boldsymbol{x}_g \sim G_\theta$
10:      Collect a mini-batch of reference samples $\boldsymbol{x}_r$ from $\mathcal{M}$
11:      Update $G_\theta$ via Eq (6)
12:    **end for**
13:    Update $\mathcal{M}$ by $G_\theta$ with reward calculated by $D(\boldsymbol{x}_g) - D(\boldsymbol{x}_r)$
14: **until** Convergence

---

## C  TRAINING AND EVALUATION DETAILS

### C.1  MODEL DETAILS

We follow most of the hyperparameters used in the benchmark platform Texygen Zhu et al. (2018). Specifically, the generator is a one layer LSTM with embedding size and hidden size 32. The discriminator is implemented as a TextCNN with a filter size of [2,3] and a filter number of [100,200]. The proposed self-adversarial learning paradigm introduces the relative weights for credit assignment when a generated sample is found to be better, indistinguishable or worse compared with another sample generated by the generator itself. We tuned it based on the performance in synthetic experiment and set $w_0 : w_2 = 1 : -0.1$.

### C.2  CHOICE & EXPLANATION OF METRICS

Note that many previous works use self-BLEU Zhu et al. (2018) as a diversity metric. However, we find that there exist problem in the official implementation of the self-BLEU metric: Only in the first time of evaluation that the reference and hypothesis come from the same "test data" (i.e. the set of generated sentences). After that, the hypothesis keeps updated but the reference remains unchanged (due to "is-first=False"), which means hypothesis and reference are not from the same "test data" any more, and thus the scores obtained under this implementation are not self-BLEU scores. To this end, we modified the implementation to make sure that the hypothesis and reference are always from the same "test data" (by simply removing the variables "self.reference" and "self.is-first") and found

that the self-BLEU (2-5) scores are always 1 when evaluating all the models. This problem is also discussed in the openreview of the RelGAN paper Nie et al. (2018).

Based on this consideration, we decide to employ backward-BLEU which is introduced in Shi et al. (2018) and can also evaluate the diversity of generated samples. The forward and backward BLEU resemble precision and recall of generated samples with respect to the test data, which measures the generation quality and the generation diversity respectively.

For BLEU metric, as there is no sentence level alignment for unconditional generation, BLEU is evaluated by using the entire test set treated as a single reference, which contains 10000 sentences. We then generate the same number of sentences as the prediction, and then compute n-gram overlap between the reference and the prediction. We did not apply brevity penalty following previous works. But we found the number of tokens generated are roughly the same across different compared models.

We briefly explain why $NLL_{gen}$ is able to measure the diversity of the generator: $NLL_{gen}$ measures the negative log-likelihood of the synthetic dataset evaluated by the generator. Intuitively, if the generator is diverse and captures more patterns in the synthetic dataset, the $NLL_{gen}$ score will be lower. In contrast, if the generator suffers from severe mode collapse and is of low diversity, the $NLL_{gen}$ will be higher as the generator fails to cover the diverse patterns in the training data.

As for the metric: $NLL_{gen}$+$NLL_{oracle}$, our motivation is that $NLL_{oracle}$ measures the best quality of the generator throughout training, while $NLL_{gen}$ measures the best diversity attained by the generator during training. However, the best quality and diversity are generally not achieved at the same time as GAN-training generally sacrifices the diversity for better quality. Therefore, we report $NLL_{gen}$+$NLL_{oracle}$ which can measure the quality-diversity trade-off, as the previous work demonstrated, as an additional reference.

### C.3 HYPERPARAMETERS

We follow most of the hyperparameters used in the benchmark platform Texygen Zhu et al. (2018). Specifically, we choose batch size to be 64, dropout keep prob to be 0.75, l2 regularization to be 0.2. We pretrain all model for 120 epochs and fine-tune them until convergence.

The proposed self-adversarial learning paradigm introduces the relative weights for credit assignment when a generated sample is found to be better, indistinguishable or worse compared with another sample generated by the generator itself. We tuned it based on the performance in synthetic experiment and find $w_0 : w_2 = 1 : -0.1$ to be a good choice for the initial weights[1] and fixed $w_1$ to be 0. We empirically find that the performance of SAL is not very sensitive to the choice of reward weights as long as the absolute value of $w_1$ is larger enough than $w_2$, which guarantees the training stability.

### C.4 HUMAN EVALUATION

Following the human evaluation setting in RelGAN Nie et al. (2018), The text quality evaluation is based on grammatical correctness and meaningfulness (i.e. if a sentence makes sense), text formatting problems (e.g., capitalization, punctuation, spelling errors, extra spaces between words and punctuations) are ignored. Detailed criteria is provided as follows:

---

[1]$w_1$ is linearly decreased to 0.8 and $w_2$ is increased to -0.2

Table 16: The human evaluation scale from 1 to 5 with corresponding criteria and example sentences.

| Scale | Criterion & Example |
|---|---|
| 5-Excellent | Its grammatically correct and makes sense. |
| | For example, "a man standing on a skateboard in the middle of the street ." |
| 4-Good | It has some small grammatical errors and mostly makes sense. |
| | For example, "two women is in a cafe look outside." |
| 3-Fair | It has major grammatical errors but the whole still conveys some meanings. |
| | For example, "a man riding on on motor scooter and window." |
| 2-Poor | It has severe grammatical errors and the whole doesn't make sense, |
| | but some parts are meaningful. |
| | For example, "a blue bike on on on a dirt bike ." |
| 1-Unacceptable | It is basically a random collection of words. |
| | For example, "a motorcycle close close on it and ." |

## C.5 ADDITIONAL RESULTS ON CAL AND LEAKGAN

In this section, we present the performance comparison on SAL vs CAL (comparative adversarial learning, which uses comparative discriminator but does not train the model with self-play.) and the application of SAL on LeakGAN. We find that SAL significantly outperforms CAL on all dataset. In addition, we find that the proposed self-adversarial learning paradigm can also be applied on other text GAN architectures, e.g. LeakGAN, and help improve its performance.

Table 17: Performance comparison of different models in synthetic tests where sequence length is set to 20 and 40 respectively. For all metrics presented, lower value is better.

| Method | $\text{NLL}_{oracle}(20)$ | $\text{NLL}_{oracle}(40)$ |
|---|---|---|
| SAL | 7.71 ±0.17 | 9.31±0.03 |
| CAL | 8.01 ±0.24 | 9.43±0.05 |
| LeakGAN | 7.04 ±0.37 | 7.19±0.35 |
| LeakGAN + SAL | **6.69** ±0.29 | **6.91**±0.27 |

Table 18: Performance comparison of different models in the COCO caption generation task. Metrics from top to bottom represent respectively the generation quality, the generation diversity, and the divergence between real data of generated sentences. For all BLEU metrics, higher value is better, for $\text{NLL}_{gen}$ and FD, lower is better.

| Metrics | CAL | SAL | LeakGAN | LeakGAN + SAL |
|---|---|---|---|---|
| BLEU-2(F) | 0.767 ±0.03 | 0.785 ±0.02 | 0.749 ±0.03 | **0.798** ±0.03 |
| BLEU-3(F) | 0.541 ±0.03 | 0.581 ±0.03 | 0.532 ±0.04 | **0.592** ±0.03 |
| BLEU-4(F) | 0.337 ±0.03 | 0.362 ±0.02 | 0.353 ±0.03 | **0.369** ±0.02 |
| BLEU-5(F) | 0.211 ±0.02 | 0.227 ±0.02 | 0.233 ±0.03 | **0.241** ±0.02 |
| Perplexity | 276.7 ±12.5 | 231.3 ±10.8 | 291.4 ±12.5 | **218.2** ±10.8 |
| BLEU-2(B) | 0.705 ±0.03 | 0.724 ±0.02 | 0.733 ±0.03 | **0.741** ±0.02 |
| BLEU-3(B) | 0.479 ±0.03 | 0.503 ±0.03 | 0.512 ±0.03 | **0.529** ±0.03 |
| BLEU-4(B) | 0.296±0.03 | 0.313±0.02 | 0.321 ±0.03 | **0.334**±0.02 |
| BLEU-5(B) | 0.191 ±0.03 | 0.198 ±0.02 | 0.206 ±0.03 | **0.211** ±0.02 |
| $\text{NLL}_{gen}$ | 0.936 ±0.03 | 0.873 ±0.02 | 0.683 ±0.03 | **0.655** ±0.02 |
| FD | 0.058 ±0.016 | 0.051 ±0.014 | 0.048 ±0.016 | **0.044** ±0.014 |

