# OpenReview forum: "Self-Adversarial Learning with Comparative Discrimination for Text Generation"
_ICLR.cc/2020/Conference — Accept (Poster)_

### Official Review · AnonReviewer1 · 2019-10-23
**Official Blind Review #1**

**Rating:** 8

**Review:**

To alleviate the issues of reward sparsity and mode collapse in most text-generation GANs with a binary discriminator, this paper proposes a self-adversarial learning (SAL) framework with a novel comparative discriminator that takes pairs of text examples from real and generated examples and outputs better, worse, or indistinguishable. Inspired by self-play in reinforcement learning, SAL employs self-play to reward the generator to generate better samples than previous samples with self-improvement signals from the comparative discriminator. It is argued that, because the comparative discriminator always produces self-improvement signals during the training and the self-improvement signal will not be very strong when generated samples are already good enough, the issues of reward sparsity and mode collapses in conventional text GANs are reduced. Experimental results on synthetic data and benchmark datasets demonstrate that SAL outperforms SeqGAN, MaliGAN, and RankGAN both quantitatively and qualitatively.

Pros:

This paper is well-written. The studied problem is well-motivated and the method is clearly presented. The SAL framework with self-play and a comparative discriminator is novel and the results are convincing.

Cons:

1) SAL has a new discriminator, which can be viewed as an architecture change. Although it has a new training strategy and is very different from recent text GANs such as LeakGAN and RelGAN, the latest results of these recent methods on the benchmark datasets should be included in Table 4 and 5 for reference.

2) The comparative discriminator is novel and well-suited for comparing pairs of samples. However, the only informative signals come from pairs of data with one from a real sample and the other from a generated sample. During the self-play process, why the signals from the comparative discriminator comparing two generated samples are always trustworthy? Do the reward signals always help train a better generator?

3) In section 4.3, since the self-play and the comparative discriminators are shown to be the most significant, it is better to clearly show the algorithms as in Algorithm 1 for training these two baselines either without self-play or without the new discriminator. Does replacing the binary classifier with WGAN classifier help here? All these details should be included in the appendix.

4) Missing training details: It is unclear how the model architectures are chosen, and learning rate, optimizer, training epochs etc. are also missing. All these training details should be included in the appendix.

In summary, this paper proposes a novel discriminator with a new training strategy suited for the discriminator for text generation. Experimental results demonstrate the effectiveness of the proposed framework. I like the idea in the paper and am happy to vote for acceptance.


**Experience Assessment:**

I have read many papers in this area.

**Review Assessment: Checking Correctness Of Derivations And Theory:**

I carefully checked the derivations and theory.

**Review Assessment: Checking Correctness Of Experiments:**

I assessed the sensibility of the experiments.

**Review Assessment: Thoroughness In Paper Reading:**

I read the paper thoroughly.

---

> ### Author Response · Authors · 2019-11-08
> **Response to Official Blind Review #1**
>
> Thanks for your supportive review and valuable feedback!
>
> Question (1): SAL has a new discriminator, which can be viewed as an architecture change. Comparison with LeakGAN and RelGAN should be included.
>
> Answer: As the core of a GAN is its generator, the "architecture" we say in our paper mainly refers to the architecture of the generator. For SAL in our paper, its generator's architecture does not change: it is the same with that used by SeqGAN, RankGAN, and MaliGAN. The only difference is its discrimination mechanism. In contrast, LeakGAN and RelGAN employ a more complicated generator, which is orthogonal to the main focus of this paper. That’s the reason why we did not compare with them. However, according to our additional experiments, we find the GANs with advanced generator like LeakGAN can also benefit from our training approach (see Table 17&18 in Appendix). We are willing to include the results in Table 4 if you think it is necessary.
>
>
> Question (2):  The comparative discriminator is novel and well-suited for comparing pairs of samples. However, the only informative signals come from pairs of data with one from a real sample and the other from a generated sample. During the self-play process, why the signals from the comparative discriminator comparing two generated samples are always trustworthy? Do the reward signals always help train a better generator?
>
> Answer:  To improve the comparative discriminator for comparing two generated samples, we introduce additional weak supervision (see section 3.1) to construct pairwise instances from the generated samples to enhance the training of the discriminator to prevent it from always assigning two generated samples as “indistinguishable”. The proposed weak supervision method selects “worse” samples from the generator during earlier epochs of MLE-pretraining when the generator did not fully converge, and select “better” samples from the later epochs when the model converged. In this way, the discriminator can be taught to compare the generated samples.
> As you commented, despite the efforts we tried our best to make to improve the discriminator, there is no guarantee that the discriminator will never make a mistake, which is the same for all the GANs. However, according to our experiments, the learning signals from our discriminator should be informative and trustworthy, which result in a better generator.
>
>
> Question (3):  In section 4.3, since the self-play and the comparative discriminators are shown to be the most significant, it is better to clearly show the algorithms as in Algorithm 1 for training these two baselines either without self-play or without the new discriminator. Does replacing the binary classifier with WGAN classifier help here? All these details should be included in the appendix.
>
> Answer: We have included these details in the section B.1 of the appendix. In addition, we did not find WGAN discriminator improves the training, we suspect that it makes the reward less stable for reinforcement learning based TextGAN training. That’s to say, with a conventional binary discriminator, the reward is scaled in [0,1], but in WGAN there is no such guarantee, which makes the training less stable.
>
>
>
> Question (4):Missing training details: It is unclear how the model architectures are chosen, and learning rate, optimizer, training epochs etc. are also missing. All these training details should be included in the appendix.
>
> Answer: We have included more training details in appendix C.1 and C.3. Thanks for pointing this out!

---

### Official Review · AnonReviewer3 · 2019-10-24
**Official Blind Review #3**

**Rating:** 8

**Review:**

Summary: This paper describes a self-adversarial method to train a GAN for text generation that circumventes the problems of mode collapse and reward sparsity. They replace the traditional binary discriminator with a comparative discriminator, which provides the generator with more frequent rewards that are not always restricted to the limited number of real training examples.

They show gains on synthetic and real generation tasks.

Strengths:

The paper is well written with adequate details about the training objectives and the learning algorithm. I appreciate the human analysis conducted by the authors. The SAL method performs better than many strong baselines consistently across 3 datasets and on various metrics.

The authors present strong ablations, that are very insightful, particularly, regarding the role played by classifying sentences as indistinguishable.

Weaknesses and Questions:

In general, some more description about the motivation of each metric would be helpful, rather than just stating that its from previous work.

How is BLEU evaluated for this text generation task? Is the entire test set treated as a single reference? Do you generate the same number of tokens as the reference and then compute n-gram overlap between the reference and the prediction? What happens to the brevity penalty of BLEU?

In Table 4, does BLEU-5(F) denote only 5-gram precision, or is it the geometric mean of 1-5 gram overlaps?

How does NLL_gen serve as a measure of diversity for the synthetic dataset?

For the human evaluation, does quality mean grammaticality? Can simple memorized sentences be scored higher?

Typos in Section 4. The authors refer to tables 17 and 18. Please fix.

**Experience Assessment:**

I have published in this field for several years.

**Review Assessment: Checking Correctness Of Derivations And Theory:**

I assessed the sensibility of the derivations and theory.

**Review Assessment: Checking Correctness Of Experiments:**

I carefully checked the experiments.

**Review Assessment: Thoroughness In Paper Reading:**

I read the paper thoroughly.

---

> ### Author Response · Authors · 2019-11-08
> **Response to Official Blind Review #3**
>
> Thanks for your supportive review and valuable feedback! We agree that details and descriptions about the motivation of each metric would be helpful. We explain them in this response and have added them in the latest uploaded version of our paper.
>
> Question: How is BLEU evaluated for this text generation task? Is the entire test set treated as a single reference? Do you generate the same number of tokens as the reference and then compute n-gram overlap between the reference and the prediction? What happens to the brevity penalty of BLEU?
>
>
> Response: We calculate corpus-level BLEU, which is a common practice for evaluating TextGANs on unconditionally text generation. The corpus-level BLEU is evaluated by using the entire test set, which contains 10000 sentences, as a single reference because there is no sentence level alignment for unconditional generation. We then generate the same number of sentences, instead of tokens, as the prediction, and then compute n-gram overlap between the reference and the prediction. We did not apply brevity penalty following previous works. But we found the number of tokens generated are roughly the same across different compared models.
>
>
>
>
> Question: In Table 4, does BLEU-5(F) denote only 5-gram precision, or is it the geometric mean of 1-5 gram overlaps?
>
> Response:  BLEU-5(F) denotes only 5-gram precision.
>
>
>
> Question: How does NLL_gen serve as a measure of diversity for the synthetic dataset?
>
> Answer: NLL_{gen} measures the negative log-likelihood of the synthetic dataset evaluated by the generator. As proposed by the previous work, a lower NLL_{gen} score indicates the generator captures more diverse patterns in the synthetic dataset, while a higher NLL_{gen} score indicates that generator suffers from mode collapse and is of low diversity. In this sense, we use NLL_gen to serve as a metric for diversity.
>
>
> Question: For the human evaluation, does quality mean grammaticality? Can simple memorized sentences be scored higher?
>
> Answer:  Details about human evaluation, which are almost identical to that in RelGAN, are presented in section C.4 in the Appendix. The text quality evaluation is based on grammatical correctness and meaningfulness (i.e. whether a sentence makes sense or not); while text formatting problems (e.g., capitalization, punctuation,spelling errors, extra spaces between words and punctuations) are ignored. As you commented, simple memorized sentences are likely to receive a higher score. In practice, however, we did not find the trained generators tend to directly copy training examples. But we think your concern is inspiring and reasonable, which inspires us to consider the necessity of using an additional metric measuring whether the generator copies the training data to supplement the human evaluation for the text generation task. Thank you very much for pointing this out!
>
> We have fixed typos and added explanation about employed metrics in our revised version of the paper (Appendix C.2). Thanks for pointing this out!

---

> > ### Comment · AnonReviewer3 · 2019-11-15
> > **Thanks for all the effort that you have put into the responses.**
> >
> > Thanks for all the effort that you have put into the responses. The new version is significantly better.
> >
> > How does NLL_gen serve as a measure of diversity for the synthetic dataset?
> >
> > -----> I still am not entirely convinced of this. It could be that the output is just bad, and has nothing to do with diversity. If we take the same words and garbled it up, the diversity is the same, but the NLL_{gen} will be really low. Is this not the case?

---

> > > ### Author Response · Authors · 2019-11-15
> > > **The explanation for NLL_gen**
> > >
> > > Thanks for your comments!
> > >
> > > NLL_{gen} evaluates the "diversity" of the generator instead of the diversity of the output, as we use the test data generated by the oracle LSTM as the input when evaluating the NLL_{gen}. Intuitively, NLL_{gen} is the loss of the generator given a test set, which captures the model’s capacity to fit the distribution of the test data. If the generator can capture most modes in the test data, the NLL_{gen} will be low; on the other hand, it will be high if the generator collapses to only limited modes. This metric is first proposed as a diversity metric in the TexyGEN paper because the generator suffering from mode collapse is unlikely to get a good NLL_{gen} score, as a supplement to the NLL_{oracle} score that measures the quality of the generator. Then, it is widely used by a lot of following work on text generation including RelGAN and COT for evaluating the diversity of the generator as it can measure how much a generator suffers from mode collapse during GAN training.
> > >
> > > We agree with you that the diversity of the generator and the NLL_{gen} score are not completely equivalent: a high NLL_{gen} does not necessarily infer that the generator is of low diversity -- as you said, the reason for a high NLL_{gen} might also be that the generator's quality is poor. Actually, a high NLL_{gen} score is a necessary but not sufficient condition for low diversity. To compare with the previous work on TextGANs, however, we follow the previous work to refer this metric as a diversity metric. We appreciate your suggestion and will make it clearer in our manuscript to make it easy and clear for readers to understand this metric.

---

### Official Review · AnonReviewer2 · 2019-10-27
**Official Blind Review #2**

**Rating:** 3

**Review:**

This paper introduces a Self-Adversarial Learning (SAL) mechanism in GAN based text generation, aiming at tackling the problem of mode collapse and sparse rewards problem. Specifically, motivated by “self-play” mechanism in RL community, instead of using a binary classifier as discriminator in original GAN, SAL employs a comparative discriminator which is a pairwise classifier with three classes: “better”, “worse” and “indistinguishable”. The authors provide lots of experimental results and ablation study showing the efficiency of the proposed mechanism in comparison with previous GAN models.

Decision: weak reject.
This paper is well motivated: clearly sparse rewards and mode collapse are two problems need to be solved in GAN based text generation, however, the following concerns prevent me from finding this paper acceptable in ICLR:
The “self-play” idea is widely used in RL. In RL, the comparison between policies can be determined directly by the game simulation results. In text generation, such comparison is more difficult to be judged. This paper appears to assume that the generated sentences are “worse” than the real samples, which is similar to the original definition of discriminator in GAN, and the generated sentences in earlier epochs are “worse” than that in later epochs, which needs further justified.
As a simple extension to GAN, I’m not convinced that the problem of mode collapse could be solved by proposed mechanism. If the generator falls into a local minimum, a collapsed mode, the proposed mechanism will never pull the generator out of that. Moreover, what is the theoretical foundation of the proposed evaluation metric on quality-diversity trade-off, NLL_{gen} + NLL_{oracle}?
The setting of the important set of hyper-parameters, reward weights, is unclear in the paper. The reward weights directly influence the reward in training, thus should play an important role of model performance. More discussion about this should be provided.
Moreover, in the paper, only comparison between proposed mechanism with GAN based models are shown. Comparison with more recent models like RelGAN should be provided. And comparison with other state-of-the-art text generation model should be discussed.



Feedbacks:
References regarding experimental results in table are incorrect. For example, the results in synthetic data should be in “Table 2” and COCO image caption dataset should be in “Table 3”.
Some imprecise parts, for example, in Equation (5) and (6), it should be G_\theta(Y_{1:t-1} and G_\theta(y_t | Y_{1:t-1} .
I’m curious about what the performance would be like if the weakly supervision by regarding sentences generated in later training stage are “better” than the sentences generated in earlier training stage is removed in training comparative discriminator. This is different from the CAL model in the ablation study.
What is the influence of different values of rewards weight?
Also about the rewards weight, in the description of Scheduled rewarding, the rewards weight is described to be linearly changed with training iteration, while in Appendix C.3, the rewards weight is described to be fixed. This is very confusing.
A minor issue: for image captioning, a lot other metrics are widely used in measuring the model performance, e.g. CIDEr, SPICE and so on. Those metrics could be helpful for audience to understand how the model performs in comparison with other captioning models.
Another minor issue: for image captioning and WMT (conditional generation), the detailed model structures are not described in paper, which is not very friendly to audiences with relatively little knowledge in related areas.



**Experience Assessment:**

I have published one or two papers in this area.

**Review Assessment: Checking Correctness Of Derivations And Theory:**

I assessed the sensibility of the derivations and theory.

**Review Assessment: Checking Correctness Of Experiments:**

I assessed the sensibility of the experiments.

**Review Assessment: Thoroughness In Paper Reading:**

I read the paper at least twice and used my best judgement in assessing the paper.

---

> ### Author Response · Authors · 2019-11-08
> **Response to Official Blind Review #2**
>
> Thanks for your valuable review and feedback!
>
> Question:  This paper appears to assume the generated sentences in earlier epochs are “worse” than that in later epochs, which needs further justified.
>
> Response:  The assumption that samples in the earlier epochs are worse than those in the later epochs is for providing supervision for the comparative discriminator to compare the quality between the generated samples for self-play training. We select the samples from the checkpoints of the first three epochs (whose perplexity on the dev set is about 30000/1000 on the synthetic and Image COCO dataset respectively) of the MLE pre-training as the “worse” samples, and select the samples from the last three epochs' checkpoints (whose perplexity is about 5000/300 on the two datasets respectively) as the “better” samples. Since the perplexity of the earlier epochs checkpoints is much larger than that of the later epochs, it is reasonable to consider the earlier samples to be worse than the later samples.
>
>
> Question: The problem of mode collapse could be solved by proposed mechanism is not convincing.
>
> Response: We agree with you that the problem of mode collapse cannot be easily “solved”. We never claim that we solved the problem but we can alleviate mode collapse with our self-play mechanism. Its intuition is that if a generator starts to generate samples from one mode (A)  more frequently than from other modes, the generated samples in mode A will more likely to be compared to other previously generated samples in the same mode, and will be assigned to be “indistinguishable” and thus receive 0 reward. In this way, the risk of mode collapse can be reduced to some extent, which has been verified by our experimental results in NLL_{gen} and Backward BLEUs in both synthetic and real datasets.
>
>
> Question: What is the theoretical foundation of the proposed evaluation metric on quality-diversity trade-off, NLL_{gen} + NLL_{oracle}?
>
> Response: We adopt the metric NLL_{gen} + NLL_{oracle} following the previous work: COT: cooperative training for generative modeling of discrete data (ICML 2019). The motivation of using NLL_{gen} + NLL_{oracle} is to consider both quality (NLL_{oracle}) and diversity (NLL_{gen}). We admit it is simple and does not have much theoretical foundation, but it can reflect both quality and diversity in a single metric, and can be used as a referential evaluation metric, as the previous work demonstrated. We included an explanation of this metric in the section C.2 of the Appendix in the latest version uploaded.
>
>
> Question: The setting of the important set of hyperparameters, reward weights, is unclear in the paper.
>
> Response: We provided details about the model architecture and hyper-parameters in Appendix C.1 and C.3. We empirically set the hyperparameters following TexyGen. We have uploaded a revised version with more details and discussions about the hyperparameters.
>
>
> Question: Comparison with more recent models like RelGAN should be provided.
>
> Response: As the main focus of our paper is the training paradigm (or objective) and does not involve modifications of the architecture of the generator, we only compare our approach with TextGANs with the same architecture but different training objectives, i.e. SeqGAN, RankGAN, and MaliGAN. We did not compare with the GANs that modify the model architecture (e.g., LeakGAN and RelGAN) because they are orthogonal to our contribution. However, according to our additional experiments, we find the GANs with advanced architecture like LeakGAN can also benefit from our training approach (see Table 17&18 in Appendix).
>
> Other Responses:
> We will fix references and equations in the revised version, thanks for pointing this out!
> The performance of our model without weak supervision is indeed comparable with the full SAL model in synthetic and COCO datasets, but worse (F-BLEU3 from 0.52 -> 0.50) on the WMT dataset. We think it is because this corpus is more challenging and discriminating real sentences from generated ones are easier, which makes the discriminator easily to be over-trained, which makes the weak supervision necessary.
> The influence of different values of rewards weight is not very significant, as long as the negative reward w2 is not very large, which stabilizes RL training.
> The rewards weight in C.3 is that in the beginning of GAN training, we have made it clear in the revision.
>
> Question: Details for image caption and machine translation.
>
> Response: We generate sentences unconditionally in COCO and WMT datasets following previous work (e.g., RankGAN, LeakGAN, RelGAN, etc.). In other words, we only use the text corpora instead of performing these tasks. (Conditional generation is possible with a conditional discriminator, but it is not the focus of this paper.)

---

### Author Response · Authors · 2019-11-08
**Latest version of manuscript uploaded**

We have uploaded the latest version of our manuscript, which addresses some concerns of the reviewers.
Changes include fixes of several typos, clarifications of model details and hyper-parameters, and detailed explanation of metrics.  We also included a more detailed description of the weak supervision approach in section 3.1.
Many thanks to the reviewers for their valuable feedback.

---

### Decision · Program_Chairs · 2019-12-19

**Decision:**

Accept (Poster)

**Comment:**

This paper proposes a method for improving training of text generation with GANs by performing discrimination between different generated examples, instead of solely between real and generated examples.

R3 and R1 appreciated the general idea, and thought that while there are still concerns, overall the paper seems to be interesting enough to warrant publication at ICLR. R2 has a rating of "weak reject", but I tend to agree with the authors that comparison with other methods that use different model architectures is orthogonal to the contribution of this paper.

In sum, I think that this paper would likely make a good contribution to ICLR and recommend acceptance.